# Feature Importance Analysis for Mini Mental Status Score Prediction in Alzheimer's Disease

**Howard Prioleau & Saurav K. Aryal**
Department of Electrical Engineering and Computer Science
Howard University
Washington, DC 20059, USA
`howard.prioleau@bison.howard.edu` & `saurav.aryal@howard.edu`

## Abstract

This research article proposes developing predictive models to forecast Mini-Mental State Exam (MMSE) scores using the 54 most important features identified from the current state-of-the-art model. The study employs the SHapley Additive exPlanations (SHAP) method to explore feature importance and interpret model performance. The analysis shows that the Automated Readability Index (ARI) is the most influential feature in predicting MMSE scores. This finding suggests that ARI's capability to capture language impairment and morphosyntax is valuable in predicting cognitive decline in dementia patients. Although the analysis could not evaluate all features, this study provides a foundation for future investigations into features that may assist in predicting MMSE scores and the onset of Dementia.

## 1 Introduction

According to the Alzheimer's Association, the prevalence of Alzheimer's Disease (AD) in adults aged 65 and above is estimated to be 1 in 9 individuals (Harun et al., 2016). Since AD can impact an individual's speech and cognitive function (Ferrer et al., 1983; Lira et al., 2014; Rudzicz et al., 2012), researchers have focused on exploring the association between acoustic-linguistic features and AD detection. While medical literature has extensively documented AD's acoustic and linguistic symptoms, their analysis through Machine Learning / Natural Language Processing (NLP) is still limited. Recent studies have compared features for AD classification (Li et al., 2021), but few have explored feature analysis for granular Mini Mental Status Exam (MMSE) prediction. To address this gap, the present study adopts the top 54 features from the current State of The Art (SOTA) (Aryal et al., 2022) approach and trains multiple machine learning models on these features. The best-performing model is then scrutinized using SHapley Additive exPlanations (SHAP) values Lundberg & Lee (2017) to identify the most critical features for predicting MMSE scores and to improve AD detection.

## 2 Methodology

This study utilizes the ADReSS Challenge dataset Luz et al. comprising of audio and transcript data from 156 patients performing the Cookie Theft task from the Boston Diagnostic Aphasia exam. We utilized the current SOTA Aryal et al. feature set which extracted 13,000+ features and they were able to utilize RFECV to get it down to the top 54 features.

We employ Light Gradient Boosted Machine (lightGBM) (Ke et al., 2017), Kernel-based Support Vector Machine (SVM), standard distance-based k-nearest neighbors (kNN), and Aryal et al.'s linear model with SGD optimization, which have been previously utilized in healthcare (Callahan & Shah, 2017). The models are fine-tuned through Bayesian Hyperparameter Optimization. The performance is evaluated using Root Mean Square Error (RMSE) and a 95% confidence interval over 20 random seeds. To gain a more comprehensive understanding of the importance and relevance of features in predicting MMSE scores, a feature analysis will be conducted using SHAPly values on

the best-performing model tested on the dataset. SHAP values will be used to rank and quantify the significance of each feature. This process will enable a thorough examination of the features that offer the most value for accurately predicting MMSE scores through regression modeling.

Feature Analysis will be performed on the best-performing model on the test dataset with SHAPly values. SHAP will be used to quantify features' importance and ranking. This will provide a deeper understanding of features that provide the most value for regression modeling MMSE scores.

## 3 RESULTS

|  | **lightGBM** | Aryal et al. | kNN | SVM |
|---|---|---|---|---|
| Train RMSE | **2.07 ± .24** | 4.47 ± .44 | 4.33 ± .47 | 4.11 ± .36 |
| Test RMSE | **2.91 ± .02** | 4.1 ± .38 | 3.59 ± .05 | 3.84 ± .14 |

Table 1: Regression Results

In Table 1, all the models trained outperformed Aryal et al.'s model, suggesting the need for further feature-level analysis. The best-performing model, LightGBM will be analyzed using SHAP and TreeExplainer Lundberg et al. (2020) to determine which features contribute to its success.

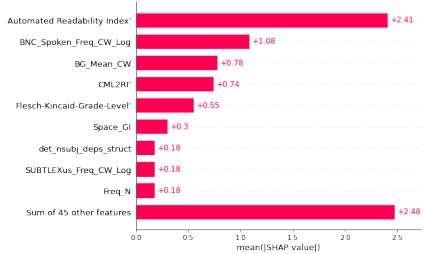
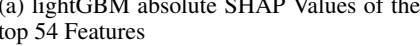

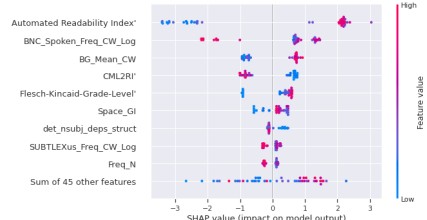

(a) lightGBM absolute SHAP Values of the top 54 Features

(b) Summary plot of the top SHAP value features and their impact of predictions

Figure 1: Plot of SHAP Values

As seen in Figure 1 the top 6 features account for most of lightGBM predictions with Automated Readability Index(ARI) having the most significant impact. Feature analysis will be done on ARI to account for the brevity requirements, and further analysis will be considered in future work.

ARI quantifies text understandability by computing the average number of words in a sentence and characters per word (Kincaid et al., 1975). Since AD negatively impacts morphosyntax, which combines morphology and syntax (Altmann et al., 2001; Taler & Phillips, 2008), poor morphosyntax results in low-complexity words and sentences are reflected in ARI. Further analysis, utilizing Spearman's (0.788) and Pearson's correlation (0.741), shows that ARI is monotonically and linearly related to MMSE. Figure 3 demonstrates the strongest overall association between ARI and MMSE compared to the other 53 features analyzed. While trends of associations of features exist with MMSE, ARI shows the highest feature importance and associations.

## 4 CONCLUSION

This study analyzes the impact of the features on MMSE prediction using SHapley Additive exPlanations (SHAP) values. The results suggest that the ARI significantly predicts MMSE scores, as it can capture language impairment frequently observed with AD. However, it is essential to note that our work's limitation includes the incomplete analysis of the entire feature set and the reliance on manually annotated transcripts that may not be readily available. Moreover, ARI may not be generalizable to non-native or non-English speakers. Future research aims to expand upon this work by developing interpretable features that can effectively capture morphosyntactic impairments leading to better models to assist medical professionals in AD diagnosis and evaluation.

ACKNOWLEDGEMENTS

We would like to acknowledge the support of Dr. Gloria Washington, Director of the Affective Biometrics Lab, and Dr. Legand Burge from the Department of Electrical Engineering and Computer Science at Howard University for providing the financial and computational resources utilized for the project. Further thanks are extended to Dr. Legand Burge for AIM-AHEAD's PRIME Fellowship, which enabled the research project. The content is solely the responsibility of the authors and does not necessarily represent the official views of the supporting organizations.

## URM STATEMENT

The authors acknowledge that all key authors of this work meet the URM criteria of ICLR 2023 Tiny Papers Track.

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

## A    APPENDIX

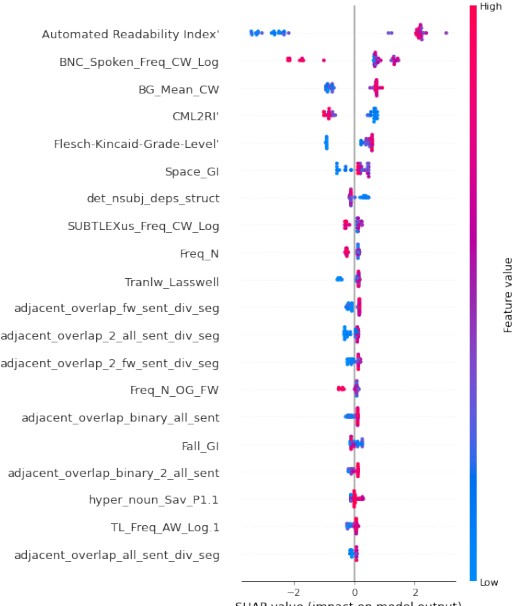

Figure 2: Expanded Impact of Features have on lightGBM prediction

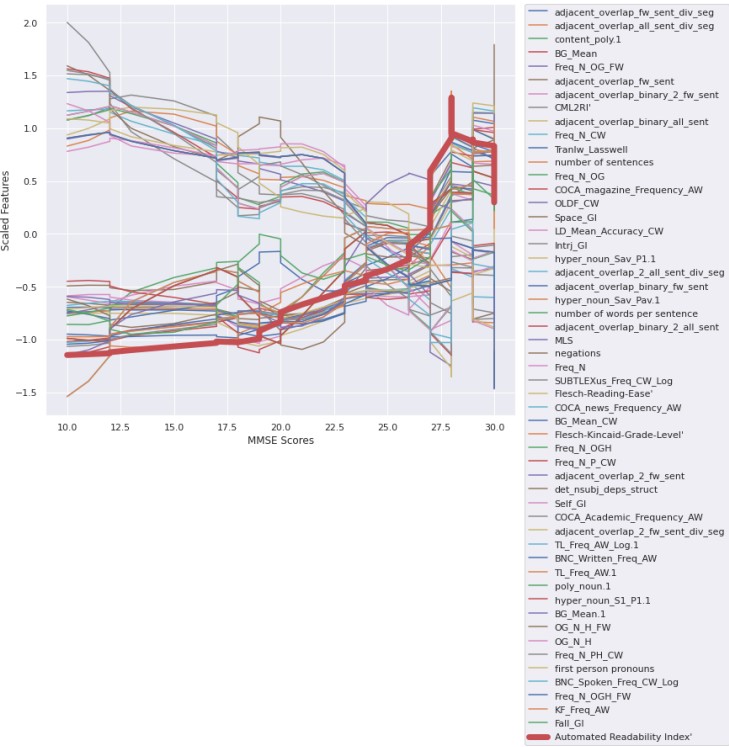

Figure 3: Plot of all 54 Features scaled for comparability as the Y and MMSE being the X. For a clearer picture of the trend of the data an Gaussian filter was applied to smooth out the data with a sigma of 2. The embolden line plot is the ARI plot.

