# OpenReview forum: "Feature Importance Analysis for Mini Mental Status Score Prediction in Alzheimer’s Disease"
_ICLR.cc/2023/TinyPapers — Submitted to Tiny Papers @ ICLR 2023_

### Official Review · Reviewer_Us3i · 2023-03-29

**Confidence:** 5

**Summary Of Contributions:**

The study employs machine learning models, to forecast Mini-Mental State Exam (MMSE) scores for Alzheimer's Disease detection. The authors then use the SHapley Additive exPlanations (SHAP) method to analyze feature importance and interpret the models' performance. The analysis reveals that the Automated Readability Index (ARI) is the most influential feature in predicting MMSE scores.

**Rating:**

Great Start (GS): a submission which meets some of the reviewing criteria but has room for improvement

**Strengths And Weaknesses:**

The authors proposed an interesting and valid problem. The study is well-organized, well-written, and addresses an important research gap in the literature. However, there are several points that need improvement:

Strengths:
1. The authors have a solid problem setting: determining the most important feature in Alzheimer's disease detection.
2. The authors conducted a sufficient literature review.
3. The authors discussed the result, stating that the Automated Readability Index (ARI) is the most influential feature, within a biological context.

Weakness:
1. The study is an incremental contribution to the problem.
2. The authors only used one comparison criterion for feature importance.
3. The features chosen for selection originate from a paper with the worst performance in comparison.

**Suggested Changes:**

1. The authors could provide more details about the feature selection process employed by Aryal et al. and the rationale behind their choices.
2. As all machine learning models tested by the authors outperform the method in the Aryal et al. paper, it may not be reasonable to use the features selected by Aryal et al.
3. A more thorough comparison would involve discussing the pros and cons of using SHAP values for feature importance analysis.
4. There are many other criteria for feature importance, such as recursive feature elimination (RFE). It would be better to compare the SHAP values to those.

---

### Official Review · Reviewer_EfuX · 2023-03-31

**Confidence:** 4

**Summary Of Contributions:**

The authors present an analysis of acoustic-linguistic features that can be used to predict Mini Mental Status Exam scores used for Alzheimer detection. Using SHAP values to explain the prediction of LGBM model they find that the ARI value best contributes to model's prediction of MMSE scores.

**Rating:**

Clear, Correct, and Reproducible (CCR): a submission which meets the reviewing criteria

**Strengths And Weaknesses:**

Strengths
- The authors propose a novel methodology to identify salient features that can be used for Alzheimer's detection.
- The paper is well written and the proposed methodology can be can be reproduced.

Challenges
- There may be a risk of authors uncovering an artifact specific to the lightGBM model as opposed to a globally salient feature that best predicts MMSE. The methodology exhibits a selection bias as only the best-performing model is selected for analysis.
- The authors should consider extending the Shapley values to other models and see if the AIR is consistently correlated with MMSE predictions.

**Suggested Changes:**

n/a

---

### Author Response · Authors · 2023-06-01
**I wish to opt-in for archival**

I wish to opt-in for archival

---

### Comment · Area_Chair_5zs2 · 2023-06-06
**Final meta-review: Invite to archive**

This work meets the threshold for archival, contents the URM statement and is deanonymized

---

### Meta-Review · Area_Chair_5zs2 · 2023-04-04

**Recommendation:** Invite to present
**Confidence:** 3

**Metareview:**

The authors present an analysis pipeline for finding which are the most salient features for predicting Mini-Mental State Exam (MMSE) scores. They find that one acoustic-linguistic feature called Automated Readability Index (ARI) is the most influential. Reviewers found the paper clear, reproducible, well organised and well-written. It is significant in that it presents a method for finding features most relevant to MMSE prediction which could assist with dementia detection.

Pros:
* Very clear explanation of the methodology.
* A solid problem setup.

Cons:
* There may be a bias since only the best performing model (out of many) was tested to determine that ARI is the best feature. There is no guarantee that ARI is the globally best feature over many models.
* The link between predicting MMSE and dementia is not well explained.
* The work incrementally builds on the work of Aryal et al. 2022. This paper, its model, how they chose the features etc. should be explained in a little more detailed.




**Summary:**

The authors present a pipeline for finding the most salient Mini-Mental State Exam (MMSE)-predicting features. Automated Readability Index (ARI) is the most influential. This could help dementia detection. The reviewers thought it was reproducible and well-written. The reviewers had an issue with the fact that the features are taken from a model which ends up performing worst in their analysis and that only the best performing model is chosen for in depth study.

**Comments And Feedback To The Authors:**

The main suggestion is to comment more on the setup by Aryal et al., how they selected their features and why their model performs poorly.

The reviewers would also suggest you to re-run the analysis on all models (not just the lightGBM, which performed best) to confirm that AREI is consistently the most salient feature, independent of model.

**Reason For Not Giving A Higher Recommendation:**

Although well done, the study is incremental in nature.

The paper convincingly asserts that ARI is the best feature from those it is selected from (i.e. those in Aryal et al.) on the model performing best in their analysis (lightGBM) but _not_ that it is the best feature overall. To receive a higher recommendation more thorough and comprehensive analysis should be performed on all features/models. Despite this it is worth noting that the two-page limit does make such an in-depth analysis hard, and the authors should be commended for what they achieve in the short space available, hence why I am recommending the second highest option ("Invite to present").

**Reason For Not Giving A Lower Recommendation:**

Most of the listed weaknesses and suggested changes by the reviewers are minor and commentary based. Overall the paper is well explained, well written and the conclusions are justified given the methodology and data shown. It is a nice piece of work and has the potential to be impactful.

---

### Decision · Program_Chairs · 2023-04-07

Invite to present